# Endometrial Cytology in Diagnosis of Endometrial Cancer: A Systematic Review and Meta-Analysis of Diagnostic Accuracy

**DOI:** 10.3390/jcm12062358

**Published:** 2023-03-17

**Authors:** Ting Wang, Ruoan Jiang, Yingsha Yao, Yaping Wang, Wu Liu, Linhua Qian, Juanqing Li, Joerg Weimer, Xiufeng Huang

**Affiliations:** 1Department of Obstetrics and Gynecology, Women’s Hospital, Zhejiang University School of Medicine, Hangzhou 310006, China; 2Department of Gynecology and Obstetrics, University Medical Center Schleswig-Holstein, 24103 Kiel, Germany; 3Zhejiang Provincial Key Laboratory of Precision Diagnosis and Therapy for Major Gynecological Diseases, Women’s Hospital, Zhejiang University School of Medicine, Hangzhou 310006, China; 4Department of Gynecology, Women’s Hospital, Zhejiang University School of Medicine, Hangzhou 310006, China

**Keywords:** endometrial cytology, endometrial cancer, screening test, diagnostic accuracy, meta-analysis

## Abstract

Background: Because the incidence of endometrial cancer has been increasing every year, it is important to identify an effective screening method for it. The endometrial cytology test (ECT) is considered to be the more acceptable technique compared to invasive endometrial sampling. Methods: The study followed the Priority Reporting Project for Systematic Evaluation and Meta-Analysis (PRISMA-DTA) protocol. This systematic rating searched EMBASE and Web of Science databases for studies on ECT for endometrial cancer from the databases’ dates of inception to 30 September 2022. All literature screening and data extraction were performed by two researchers, while the methodological quality of the included studies was assessed against defined inclusion criteria. And a third researcher resolves the disagreements. Results: Twenty-six studies were eventually included in this final analysis. Meta-analysis results showed that the diagnostic accuracy characteristics of ECT for endometrial cancer were as follows: combined sensitivity = 0.84 [95% confidence interval (CI) (0.83–0.86)], combined specificity = 0.98 [95% CI (0.98–0.98)], combined positive likelihood ratio = 34.65 [95% CI (20.90–57.45)], combined negative likelihood ratio = 0.21 [95% CI (0.15–0.30)], and area under the summary receiver operating characteristic curve = 0.9673. Conclusions: ECT had the ability to detect endometrial cancer with strong specificity, although some studies have demonstrated significant differences in sensitivity.

## 1. Introduction

Endometrial cancer (EC) is the most common gynecological cancer in high-income countries. Its incidence is rising globally, with 417,000 new diagnoses made in 2020 [1,2]. EC usually presents at an early stage with postmenopausal bleeding [3]. Around 15% of all diagnoses are made pre-menopause [4]. The disease burden is expected to increase due to the aging of the general population and an overall increase in the prevalence of obesity [1].

Currently, the widely used diagnostic methods for EC include ultrasonography for preliminary assessment and histological examination for definitive diagnosis. Ultrasonography is the most commonly used noninvasive method to evaluate suspected endometrial lesions, mainly based on endometrial thickness. In premenopausal women, endometrial thickness fluctuates physiologically due to periodic hyperplasia of menstruation, and other benign pathologies of the uterus, such as polyps and fibroids, may also lead to endometrial thickening under ultrasound [5]. Therefore, the discrimination ability of EC in premenopausal women based on transvaginal ultrasonography is inferior to that in postmenopausal women. Other screening methods for endometrial lesions include other imaging examinations (such as magnetic resonance imaging and CT), but the above techniques are of limited value in the screening of endometrial proliferative lesions [6,7].

EC diagnosis relies on histological examination of an endometrial tissue sample. However, this invasive test is reserved for patients with endometrial pathology or a thickened endometrium on a transvaginal ultrasound scan [5]. Although EC can be precisely classified into four molecular types, which can guide its treatment management and predict prognosis, biopsy material availability is still important. The establishment of a screening program for EC is necessary for both the general population and specific high-risk groups. Endometrial cytology test (ECT) is considered to be the more acceptable screening method compared to invasive endometrial sampling [8].

ECT is a cytological evaluation of obtained sample endometrial cells. Endometrial cytology samplings include serrated, brushed, and negative-pressure endometrial cell specimens. The specific tools include but are not limited to Li brush, Tao brush, SAP-1, Uterobrush and others. Endometrial cytology sample preparation technology refers to conventional and liquid-based cytology (LBC). Sample standardization is one of the causes of diagnostic problems that have been addressed via LBC technology [9,10]. The standardized diagnostic algorithm using cytomorphological assessment of LBC samples, also known as the Osaki Study Group method, has recently been demonstrated to be reliable and reproducible [11]. The application of immunocytochemistry and molecular technology techniques for endometrial cytology sample processing has been studied to improve the diagnostic accuracy of endometrial cytology [12,13,14,15,16,17,18]. Endometrial biopsy mainly includes D&C, hysteroscopy and hysterectomy. Routine pathological examination of endometrial tissue obtained by any sampling method is the gold standard for the diagnosis of endometrial lesions.

Because previous studies have been inconsistent on ECT methods (swab type) and the diagnostic ability of EC obtained, the aim of the present study was to conduct a systematic review and meta-analyses of published literature to determine ECT accuracy in detecting endometrial carcinoma and its relationship with cell sampling methods and reference standards.

## 2. Methods

### 2.1. Search Strategy and Study Selection

The study followed the Priority Reporting Project for Systematic Evaluation and Meta-Analysis (PRISMA-DTA) protocol [19]. EMBASE and Web of Science databases were searched for studies published from their inception to 30 September 2022.

The research question was developed using the PICOS strategy as follows: the population (P) was defined as women in pre- and post-menopause; the intervention (I) was endometrial sampling cytology as the diagnostic method; the comparison group (C) corresponded to pathological examination as the diagnostic method; the outcome (O) was defined as a diagnosis of normal endometrium or endometrial lesions (endometrial polyp, hyperplasia, and cancer); and the study design (S) included randomized controlled trial (RCT), prospective, and retrospective cohort studies. This review had no year or language restrictions. Articles not meeting the research purpose were excluded.

### 2.2. Data Extraction and Quality Control

The work of checking titles and abstracts from the electronic search was done independently by two reviewers (Ting Wang and Ruoan Jiang). Complete manuscripts of all citations that may meet the predefined inclusion criteria have been obtained. Final inclusion or exclusion decisions are made after the examination of the complete manuscript. In case of duplication, choose the latest and complete version. Any disagreements regarding inclusion are resolved by the third reviewer (Yingsha Yao) through discussion or arbitration.

This review focuses on diagnostic studies in which histological results from endometrial sampling are compared with those from reference criteria. The article requirements included studies on pre- and post-menopausal women, endometrial sampling (cytology) as the diagnostic test of interest, endometrial histological findings from (blind) D&C as the reference standard, hysteroscopy or hysterectomy with histology results obtained by targeted biopsy or D&C, and hysteroscopy or hysterectomy.

The quality of the literature was evaluated using the Cochrane risk of bias tool [20,21]. Two reviewers (Wang Ting and Jiang Ruoan) independently screened the literature, extracted the data, and evaluated the quality of the literature according to the inclusion and exclusion criteria. Data extraction forms include basic literature information, patient characteristics, and sample number. Any disagreements regarding inclusion are resolved by the third reviewer (Yingsha Yao) through discussion or arbitration. Only the studies scoring as “low” on all three items of concern for applicability were included in the review. All applicable studies on this subject were included.

### 2.3. Statistical Analysis

Meta-analysis was performed using Review Manager 5.3 software (Cochrane Library). The meta-analysis used either fixed effects or random effects models. The chi-square test was used to evaluate the statistical heterogeneity of the results and expressed as an I^2^ index. If the heterogeneity test result I^2^ is less than 50%, it is considered that there is no significant heterogeneity among the included studies, and the fixed-effect model is adopted. When heterogeneity is detected, researchers drill down to find a possible explanation. If reasonable causes are found, a subgroup analysis is performed. In addition, we also used a random effects model. *p* < 0.05 was considered statistically significant.

The combined sensitivity, combined specificity, combined positive likelihood ratio, combined negative likelihood ratio, combined diagnostic odds ratio and the corresponding 95% CI of ECT for endometrial cancer were calculated to obtain the diagnostic value of ECT for endometrial cancer. We also plotted overall receiver operating characteristic curves (SROC) for different subgroups.

## 3. Results

### 3.1. Study Selection

EMBASE and Web of Science databases were searched for studies published from their inception to 30 September 2022. At the beginning of the study, we identified a total of 11,865 relevant articles based on the search strategy. After the two researchers independently conducted comprehensive reading and screening of research data, a total of 26 studies [11,22,23,24,25,26,27,28,29,30,31,32,33,34,35,36,37,38,39,40,41,42,43,44,45,46] were selected for the final investigation. The literature screening process and results are shown in Figure 1.

### 3.2. Basic Characteristics and Quality of Selected Literature

The quality of the literature was evaluated using the Cochrane risk of bias tool [20,21]. Two reviewers independently screened the literature, extracted the data, and evaluated the quality of the literature according to the inclusion and exclusion criteria. Quality assessment of four studies (15.4%) showed a ‘low’ risk of bias for all four items (not shown), while the other 22 studies had an “unclear” risk of bias in at least one criterion in the description of methods for patient selection, index, reference test, flow, and timing. All studies scored “low” on the three items of applicability. The results of the evaluation of literature quality are shown in Figure 2. After the study selection and quality assessment, 26 articles were included in a systematic review. The basic characteristics of the included studies are shown in Table 1.

### 3.3. Diagnostic Accuracy of ECT

The chi-square test was used to evaluate the statistical heterogeneity of the results and expressed as an I^2^ index; I^2^ was more than 50%. Due to the strong heterogeneity of the final selection of studies, the random-effects model was used to summarize and analyze the results of each study. Data on sensitivity, specificity, and likelihood ratios for the diagnosis of EC were extracted from 26 articles (Table 2). The sensitivity of ECT was 98% in all 26 studies (LR- of 0.22).

#### 3.3.1. Diagnostic Accuracy of ECT Using Different Reference Standards

Data on sensitivity, specificity, and likelihood ratios for the diagnosis of EC were extracted from 26 articles (Table 2). The sensitivity of endometrial sampling using blind D&C was 81% in all 13 studies (LR- of 0.22). The sensitivity of endometrial sampling using hysteroscopy with histology was 73% in two studies (LR- of 0.29). The sensitivity of endometrial sampling using hysterectomy with histology was 89% in five studies (LR- of 0.18). The remaining six studies using nonuniform endometrial samples with histology as a reference standard (Shulan Lv presented the results of two endometrial cell sampling tools simultaneously in one article) had a pooled sensitivity of 67% and LR- of 0.31. The specificity was 98–99%, regardless of the reference standard used. Figure 3A shows the SROC plot of the performance of 26 studies that calculated both sensitivity and specificity.

#### 3.3.2. Diagnostic Accuracy of ECT with Different Cytological Sampling Methods

Data on sensitivity, specificity, and likelihood ratios for the diagnosis of EC were extracted from 26 articles (Table 3). The sensitivity of endometrial sampling was 96% in three studies using Li brush (LR- of 0.06). The sensitivity of endometrial sampling was 73% in four studies using Tao brush (LR- of 0.30). The sensitivity of endometrial sampling was 84% in five studies using SAP-1 endometrial cytology collector (LR- of 0.34). The sensitivity of endometrial sampling was 99% in three studies using Uterobrush (LR- of 0.48). The sensitivity of endometrial sampling was 88% in two studies using other cytological collection methods (LR- of 0.16). Specificity was 96–99% regardless of the reference standard used. Figure 3B shows the SROC plot of the performance of the 26 studies that calculated both sensitivity and specificity.

## 4. Discussion

Because previous studies have been inconsistent on ECT methods (swab type) and the diagnostic ability of EC obtained, the present study was to conduct a systematic review and meta-analyses of published literature. The present meta-analysis assessed the diagnostic accuracy of endometrial cytology for the diagnoses of EC and compared it to different reference standards and cytological sampling methods. The specificity of endometrial sampling was very high regardless of the sampling method type or the reference test used. On the other hand, the sensitivity of different cell sampling methods was quite different, especially for Uterobrush. A benign result was not always indicative of a negative EC diagnosis. Therefore, it could be concluded that a positive test result for endometrial sampling was very accurate in diagnosing EC. However, ECT was not very accurate in ruling out EC. Thus, further diagnostic work-up for focal pathology was warranted after a benign ECT result. This is of particular importance when considering the molecular classes of EC.

The current practice is to exclude the possibility of malignant disease using sequential transvaginal ultrasound scans, outpatient hysteroscopy, and endometrial biopsies. However, this diagnostic strategy has limitations [47]. Failed outpatient hysteroscopy and endometrial biopsy, mostly due to technical failure or pain intolerance, affect up to 31% of women [48], demanding repeat investigations under general anesthesia [49]. Universal endometrial sampling on a periodic basis is probably impractical and may be unnecessary. However, patients at high risk for EC require frequent periodic screening [37]. A simple, easy-to-administer, non-invasive test that can triage women with EC for diagnostic testing while safely reassuring the vast majority of healthy women would improve patient care [29]. ECT provides a solution for obtaining an outpatient endometrial cytology sample without anesthesia, but sampling insufficiency and costs remain problems that need to be solved for the effective screening of endometrial lesions [50]. Just as D&C can miss focal pathology, ECT can also miss EC in an endometrial polyp [49]. This suggests that ECT, especially brush sampling, which is performed as a mini-curettage, can also miss a significant number of focal pathologies and possibly focal cancers. Therefore, D&C has been almost completely replaced by hysteroscopy as the reference standard, both in clinical as well as in research settings [51]. Given the above findings, further diagnostic work-up for focal intracavitary pathology in patients with a failed, insufficient, or benign ECT result seems warranted.

Endometrial biopsy mainly includes D&C, hysteroscopy and hysterectomy. Routine pathological examination of endometrial tissue obtained by any sampling method is the gold standard for the diagnosis of endometrial lesions. However, there are some differences in the omission of endometrial lesions by different methods of histological acquisition, which is mainly due to the characteristics of the technique itself: D&C is performed visually, which is greatly affected by the operator’s experience, and the tools are metal tools, which are greatly affected by the uterine cavity shape. Because hysteroscopy can be targeted under the microscope, the examination is more comprehensive, for scratching difficult tissue can be assisted by other tools; Hysterectomy is the most comprehensive examination of endometrial tissue. Because the uterine cavity shape is not completely regular, the uterine shape often presents a forward or backward tilt state, different sampling tools due to the design of the brush head differences in shape and material differences, brush rod material and bendability differences, can cause different brush tools to affect the ECT results. Although the present study found significant differences in the sampling sensitivity of different sampling brushes or different reference standards, meta-regression results showed that brush type was not the source of heterogeneity (*p* > 0.05). We still conducted a subgroup analysis of the main factors that could theoretically influence the diagnostic efficacy of ECT, including ECT sampling tools and pathological tissue sampling method. Previous studies have shown no significant difference in the diagnostic ability of EC based on different brush-sampling ECT results [34]. Tools for cytological sampling can be selected based on the patient’s economic considerations and the hospital’s capabilities [52].

Both cytology and histology are highly dependent on the skill of the pathologist [53]. Although all studies suggest that the participation of senior pathologists is preferable, the heterogeneity caused by this factor cannot be ignored or avoided. Due to the limited potential of the cytological examination for the detection of endometrial carcinoma and the rapid development of artificial intelligence in science and technology, using equipment to partially replace manual screening can improve the stability and efficiency of currently available technology [54,55,56].

An important strength of the present meta-analysis was the thoroughness of the performed search for articles on the diagnostic accuracy of ECT in pre- and postmenopausal women with and without symptoms. Because the search was carried out using all synonyms for ECT, it is likely that all articles on the subject were included in the analysis. In addition, all eligible articles were evaluated regardless of the language used. We also conducted a subgroup analysis of the main factors that could theoretically influence the diagnostic efficacy of ECT, including ECT sampling tools and pathological tissue sampling method.

This study has some limitations. Any systematic review would be concerned with the risk of publication bias and omission of potentially relevant articles. Meta-regression was used for the tissue sampling method, study area, and study implementation method, which were not considered to be the main sources of heterogeneity in the present study (*p* > 0.05). We mitigated the problem in a number of ways, including using robust search strategies, checking cross-references, and consulting with clinical librarians. Finally, because only a small number of studies were available and some of them were based on small sample sizes, conclusions had to be drawn based on a limited number of patients. Questions of strong clinical relevance, including the detection efficacy of ECT technique before and after menopause and diagnostic comparison with ultrasound, the above questions have not been solved in this paper because of the lack of the original literature.

The results of this systematic review suggested that ECT had the ability to detect EC with strong specificity, although significant variations in sensitivity were present among a number of prior studies. Although heterogeneity in the literature was not found to be directly attributable to differences in sampling tools, no brush is currently recommended for widespread use, and more research is needed to determine which form of endometrial cytology collection can be used to achieve higher detection capacity and less patient suffering. For tumor-related studies, more and more attention has been paid to genomics and metabolomics, while ECT sampling can obtain the matrix around cells. When the information provided at the cellular level is insufficient, these matrices can provide other relevant information to the clinic to improve the accuracy of diagnosis and reduce the financial and physical burden of repeated examination for patients. In order to improve the accuracy and stability of cytological testing, searching for laboratory test indicators for combined screening strategies in addition to the use of a computer should be considered. We know that the diagnostic accuracy of the test is closely related to the prevalence of the disease in which it is used. In the future, we need to conduct more large-sample (prospective) cohort studies to explore the accuracy and cost-effectiveness of ECT in the diagnosis of endometrial lesions.

## Figures and Tables

**Figure 1 jcm-12-02358-f001:**
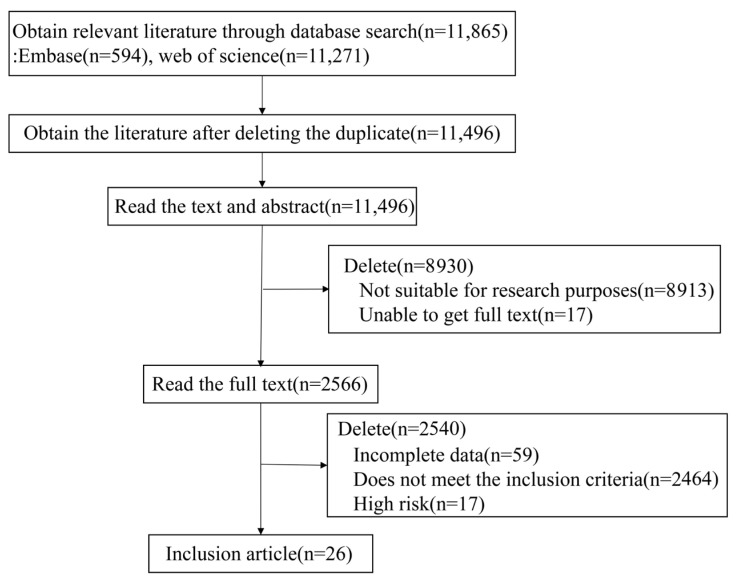
Literature screening process and flow.

**Figure 2 jcm-12-02358-f002:**
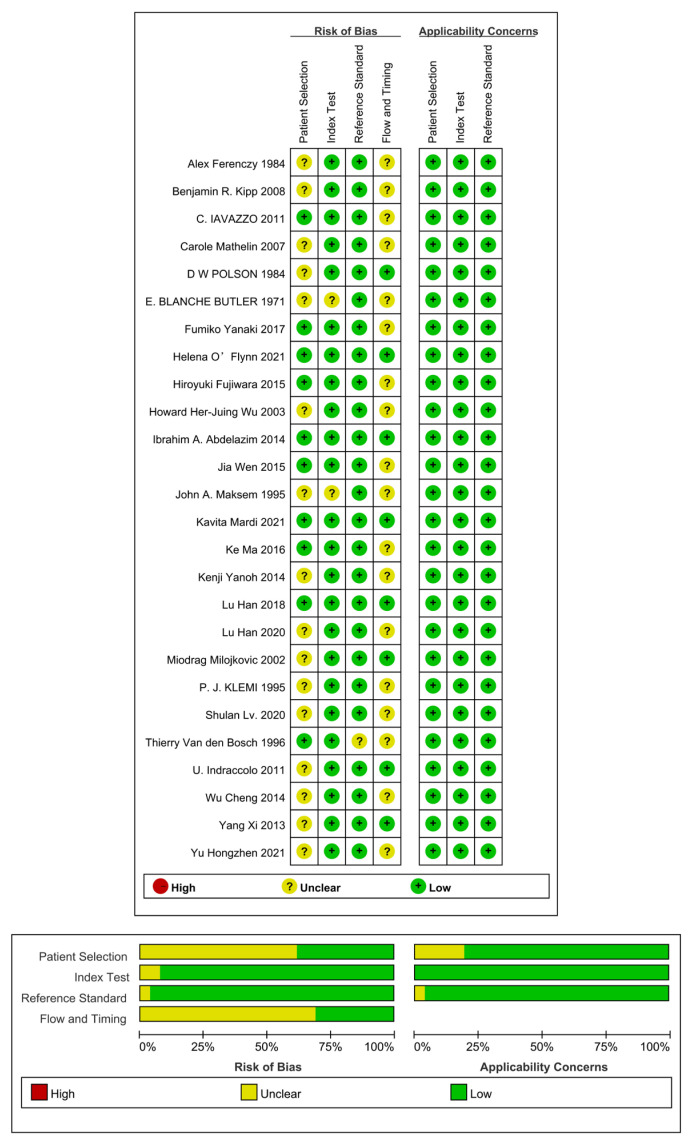
QUADAS-2 quality of selected literature [11,12,23,24,25,26,27,28,29,30,31,32,33,34,35,36,37,38,39,40,41,42,43,44,45,46].

**Figure 3 jcm-12-02358-f003:**
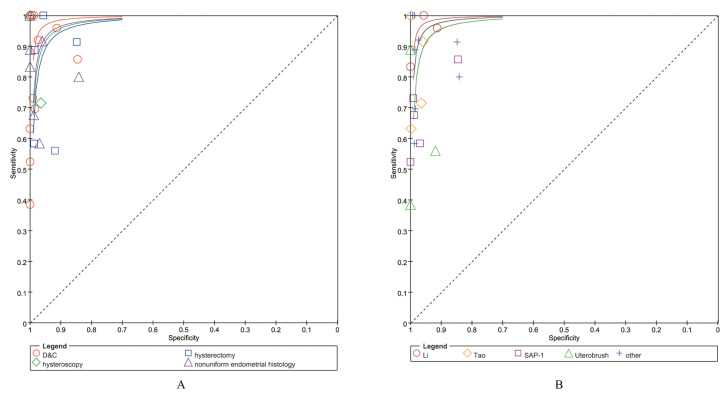
SROC (summary receiver operating curve) plot ((**A**) with different reference standards; (**B**) with different cytologic sampling methods).

**Table 1 jcm-12-02358-t001:** Study characteristics of the 26 included studies.

	Country	Study Design	ECT Sampling	Reference Standard	Menopausal Status	Effective Sample Number
Alex Ferenczy, 1984 [25]	Canada	prospectively	Endocyte	D&C	pre-&post-menopause	180
Benjamin R. Kipp, 2008 [32]	USA	prospectively	Tao brush	Hysterectomy	pre-&post-menopause	139
C. Iavazzo, 2011 [30]	Greece	prospectively	Uterobrush	Histopathologic findings	pre-&post-menopause	100
Carole Mathelin, 2007 [38]	France	prospectively	Endobrush	D&C	not clear	150
D. W. Polson, 1984 [41]	UK	prospectively	Aspiration	Curettage	postmenopause	51
E. Blanche Butler, 1971 [23]	UK	prospectively	Kuper brush	D&C or hysterectomy	pre-&post-menopause	41
Fumiko Yanaki, 2017 [46]	Japan	retrospective	Honest Super brush	D&C	not clear	1116
Helena O’Flynn, 2021 [40]	UK	prospectively	Evalyn brush	Hysterectomy	pre-&post-menopause	216
Hiroyuki Fujiwara, 2015 [26]	Japan	prospectively	endometrial cytology	Hysterectomy	pre-&post-menopause	2802
Howard Her-Juing Wu, 2003 [44]	USA	prospectively	Tao brush	D&C	pre-&post-menopause	156
Ibrahim A. Abdelazim, 2014 [22]	Egypt	prospectively	Tao brush	D&C	pre-&post-menopause	217
Jia Wen, 2015 [43]	China	prospectively	SAP-1	D&C	pre-&post-menopause	254
John A. Maksem, 1995 [36]	USA	prospectively	MedScand cytobrush	Hysterectomy	not clear	639
Kavita Mardi, 2017 [37]	India	prospectively	aspiration cytology	D&C	postmenopause	86
Ke Ma, 2016 [35]	China	prospectively	SAP-1	D&C or hysterectomy	postmenopause	567
Kenji Yanoh, 2014 [11]	Japan	prospectively	SurePath	Endometrial biopsy or curettage	pre-/postmenopause	102
Lu Han, 2018 [28]	China	prospectively	Li brush	Hysterectomy	pre-/postmenopause	299
Lu Han, 2020 [27]	China	prospectively	Li brush	D&C	pre-/postmenopause	212
Miodrag Milojkovic, 2002 [39]	Croatia	prospectively	Uterobrush	Hysterectomy	postmenopause	62
P. J. Klemi, 1995 [33]	Finland	prospectively	Uterobrush	D&C	pre-/postmenopause	313
Shulan Lv, 2020 [34]	China	prospectively	Tao brush/Li brush	D&C or hysterectomy	pre-&post-menopause	224
Thierry Van den Bosch, 1996 [42]	South Africa	prospectively	Endopap	Hysteroscopy or hysterectomy	postmenopause	87
U. Indraccolo, 2011 [31]	Italy	prospectively	sterile brush	Hysteroscopy	not clear	37
Wu Cheng, 2014 [24]	China	prospectively	SAP-1	D&C	postmenopause	231
Yang Xi, 2013 [45]	China	prospectively	SAP-1	D&C or hysterectomy	pre-&post-menopause	1520
Yu Hongzhen, 2021 [29]	China	prospectively	SAP-1	D&C	not clear	1725

**Table 2 jcm-12-02358-t002:** Diagnostic accuracy of ECT using different reference standards.

Study and Reference Standard	Sens	Spec	LR-	LR+
**D&C**				
D. W. Polson, 1984 [41]	1.00	1.00	0.05	80.18
Ibrahim A. Abdelazim, 2014 [22]	1.00	1.00	0.17	∞
Lu Han, 2020 [27]	0.96	0.91	0.05	11.03
Jia Wen, 2015 [43]	0.73	0.99	0.27	83.31
P. J. Klemi, 1995 [33]	0.38	1.00	0.61	∞
Howard Her-Juing Wu, 2003 [44]	0.63	1.00	0.38	∞
Carole Mathelin, 2007 [38]	1.00	0.99	0.13	86.33
Kenji Yanoh, 2014 [11]	0.92	0.97	0.08	35.42
Fumiko Yanaki, 2017 [46]	0.70	0.98	0.31	43.42
Alex Ferenczy, 1984 [25]	1.00	1.00	0.17	∞
Kavita Mardi, 2017 [37]	1.00	0.99	0.05	49.4
Yu Hongzhen, 2021 [29]	0.52	1.00	0.48	∞
Wu Cheng, 2014 [24]	0.86	0.85	0.17	5.56
Pooled sens/spec/LR (range)	0.81 (0.77–0.85)	0.99 (0.98–0.99)	0.22 (0.13–0.36)	59.74 (25.17–∞)
I^2^ (%)	80.6	93.3	77.4	86.6
**Hysteroscopy**				
U. Indraccolo, 2011 [31]	1.00	1.00	0.10	61.20
Benjamin R. Kipp, 2008 [32]	0.72	0.97	0.29	20.77
Pooled sens/spec/LR (range)	0.73 (0.62–0.82)	0.98 (0.92–1.00)	0.29 (0.20–0.41)	25.68 (7.53–87.58)
I^2^ (%)	61.5	45.3	0.0	0.0
**Hysterectomy**				
Miodrag Milojkovic, 2002 [39]	0.56	0.92	0.48	6.91
Lu Han, 2018 [28]	1.00	0.96	0.01	21.93
Hiroyuki Fujiwara, 2015 [26]	0.89	0.99	0.11	60.40
John A. Maksem, 1995 [36]	0.58	0.99	0.42	45.72
Helena O’Flynn, 2021 [40]	0.91	0.85	0.10	6.02
Pooled sens/spec/LR (range)	0.89 (0.87–0.90)	0.98 (0.97–0.98)	0.18 (0.08–0.42)	19.40 (6.09–61.80)
I^2^ (%)	88.8	92.4	92.5	94.7
**Nonuniform endometrial samples**				
Shulan Lv, 2020 (1) [34]	0.92	0.96	0.09	23.15
Shulan Lv, 2020 (2) [34]	0.83	1.00	0.19	∞
Thierry Van den Bosch, 1996 [42]	0.80	0.84	0.24	5.05
E. Blanche Butler, 1971 [23]	1.00	1.00	0.07	66.86
Ke Ma, 2016 [35]	0.58	0.97	0.43	18.94
C. Iavazzo, 2011 [30]	0.89	1.00	0.15	∞
Yang Xi, 2013 [45]	0.68	0.99	0.33	56.28
Pooled sens/spec/LR (range)	0.67 (0.61–0.73)	0.98 (0.97–0.98)	0.31 (0.22–0.44)	28.63 (11.18–73.36)
I^2^ (%)	64.7	88.2	42.7	84.8
Total pooled sens/spec/LR (range)	0.84 (0.83–0.86)	0.98 (0.98–0.98)	0.21 (0.15–0.30)	34.65 (20.90–57.45)
Total I^2^ (%)	86.8	91.2	88.8	87.8

**Table 3 jcm-12-02358-t003:** Diagnostic accuracy of ECT with different cytologic sampling methods.

Study and Cytologic Sampling Method	Sens	Spec	LR-	LR+
**Li brush**				
Shulan Lv, 2020 (2) [34]	0.83	1.00	0.19	∞
Lu Han, 2018 [28]	1.00	0.96	0.01	21.93
Lu Han, 2020 [27]	0.96	0.91	0.05	11.03
Pooled sens/spec/LR (range)	0.96 (0.92–0.99)	0.96 (0.93–0.97)	0.06 (0.01–0.28)	18.61 (8.25–42.01)
I^2^ (%)	69.5	84.5	70.8	63.5
**Tao brush**				
Shulan Lv, 2020 (1) [34]	0.92	0.96	0.09	23.15
Ibrahim A. Abdelazim, 2014 [22]	1.00	1.00	0.17	∞
Benjamin R. Kipp, 2008 [32]	0.72	0.97	0.29	20.77
Howard Her-Juing Wu, 2003 [44]	0.63	1.00	0.38	∞
Pooled sens/spec/LR (range)	0.73 (0.64–0.81)	0.99 (0.97–1.00)	0.30 (0.23–0.41)	40.71 (13.37–∞)
I^2^ (%)	38.4	78.9	0.0	41.2
**SAP-1**				
Jia Wen, 2015 [43]	1.00	0.99	0.27	83.31
Ke Ma, 2016 [35]	0.92	0.97	0.43	18.94
Yu Hongzhen, 2021 [29]	0.83	1.00	0.48	∞
Wu Cheng, 2014 [24]	0.56	0.85	0.17	5.56
Yang Xi, 2013 [45]	1.00	0.99	0.33	56.28
Pooled sens/spec/LR (range)	0.84 (0.83–0.86)	0.98 (0.98–0.99)	0.34 (0.26–0.46)	40.22 (11.09–∞)
I^2^ (%)	86.8	97.1	64.2	94.7
**Uterobrush**				
Miodrag Milojkovic, 2002 [39]	0.56	0.92	0.48	6.91
P. J. Klemi, 1995 [33]	0.38	1.00	0.61	∞
C. Iavazzo, 2011 [30]	0.89	1.00	0.15	∞
Pooled sens/spec/LR (range)	0.57 (0.42–0.72)	0.99 (0.98–1.00)	0.48 (0.29–0.79)	48.59 (3.67–∞)
I^2^ (%)	67.8	86.6	53.3	75.4
**Other**				
D. W. Polson, 1984 [41]	1.00	1.00	0.05	80.18
U. Indraccolo, 2011 [31]	1.00	1.00	0.10	61.20
Carole Mathelin, 2007 [38]	1.00	0.99	0.13	86.33
Thierry Van den Bosch, 1996 [42]	0.80	0.84	0.24	5.05
Hiroyuki Fujiwara, 2015 [26]	0.89	0.99	0.11	60.40
Kenji Yanoh, 2014 [11]	0.92	0.97	0.08	35.42
E. Blanche Butler, 1971 [23]	1.00	1.00	0.07	66.86
John A. Maksem, 1995 [36]	0.58	0.99	0.42	45.72
Fumiko Yanaki, 2017 [46]	0.70	0.98	0.31	43.42
Alex Ferenczy, 1984 [25]	1.00	1.00	0.17	∞
Kavita Mardi, 2017 [37]	1.00	0.99	0.05	49.40
Helena O’Flynn, 2021 [40]	0.91	0.85	0.10	6.02
Pooled sens/spec/LR (range)	0.88 (0.87–0.90)	0.98 (0.97–0.98)	0.16 (0.10–0.26)	35.69 (15.47–92.33)
I^2^ (%)	64.8	88.2	69.6	89.9
Total pooled sens/spec/LR (range)	0.84 (0.83–0.86)	0.98 (0.98–0.98)	0.21 (0.15–0.30)	34.65 (20.90–57.45)
Total I^2^ (%)	86.8	91.2	88.8	87.8

## Data Availability

The original contributions presented in the study are included in the article; further inquiries can be directed to the corresponding author/s.

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
