# Peer review of "Endometrial Cytology in Diagnosis of Endometrial Cancer: A Systematic Review and Meta-Analysis of Diagnostic Accuracy"

_jcm, 2023, doi:10.3390/jcm12062358_

Round 1

Reviewer 1 Report

Wang et al have performed a systematic review of literature findings to determine whether the endometrial cytology test (ECT) is an effective tool for diagnosis of endometrial cancer as an alternative to the more invasive endometrial biopsy sampling technique which is the current gold standard. Many studies were considered, but 26 were eventually chosen for this analysis. The authors considered pre- and post-menopausal women, as current non-evasive diagnostic methods of ultrasonography is not as effective in pre-menopausal women. The conclusions of the article stated that though the specificity of the ECT was high, the sensitivity was lower, and called for any negative result to be followed up with a more invasive diagnostic technique. This, therefore, indicates that ECT isn't able to help reduce the burden for high risk individuals who need frequent sampling as they will still require biopsy sampling with negative ECT results. 

Major points:

1. Ultrasonography is the most widely used preliminary assessment of EC, however the authors do not do a comparison to determine whether ETC was able to diagnose EC with similar or better sensitivity and specificities to ultrasonography. If the use of ETC is being considered as an alternative to normal standard practices, it would be beneficial to add in a section comparing the ETC and ultrasound as preliminary assessment of patients. This may help to show that ETC can be useful as a diagnostic tool.

2. The authors introduced the differences in diagnostic capabilities of ultrasound scans between pre- and post-menopausal women. However this comparison in ETC sensitivities are not discussed. Perhaps the lower sensitivity of ETC is due to using pre-menopausal women in the group. This comparison would benefit from stratifying their analysis based on pre- and post-menopausal women to determine if sensitivity is better depending on menopause. In addition, If the authors can show that ETC is more sensitive in pre-menopausal women than ultrasonography, this would increase the impact factor.

3. The discussion has a paragraph on evaluating metabolites for diagnostic purposes that seems very much out of place for this manuscript. There is no introduction on current research on metabolites that are associated with EC progression and risk, and no information on if there is any biomarkers currently being used or evaluated for diagnostic purposes. Finally, these ETC tools do not measure metabolites in the serum or vaginal fluid, therefore there is no relevance for this topic of discussion. This paragraph should be removed, or added in as a single sentence in the discussion when referencing other diagnostic possibilities such as using AI. 

Minor points:

1. Some minor grammar edits needed.

2. A comparison table showing specificities and sensitivities for each of the combinations of ECT sampling swab plus the reference standards used, would be good as a supplementary analysis to determine if ECT swabs with lower sensitivity (ie. Tao Brush) works better with D&C or other reference standards.

3. Though providing a lot of information, the tables are very busy and do not visually indicate the main point the authors are trying to convey. Some extra formatting and colouring may help. Adding lines to separate the sections around the different samples types and adding bolding to the summary from each sample type may help the reader to focus on the main points.

Author Response

First, thank you very much for your review comments. Regarding your questions, we provide the following feedback:

The results of our study did show that the ECT results had problems of low sensitivity and high heterogeneity among different studies. Therefore, we focused on how to achieve sensitivity in the discussion (including AI technology to reduce the difference between pathologists, combined metabolomics analysis, etc.). Moreover, due to its high accuracy, ECT is still a good screening method for patients with low pain tolerance.

Reply to main questions:

  1. Thank you very much for your proposed comparative study on the screening of endometrial cancer between ultrasound technology and ECT technology.

Previous studies have suggested that ultrasound is only moderately accurate for endometrial cancer (especially in premenopausal women). We will consider conducting a meta-analysis of the diagnostic efficacy of the above two technologies for EC in future studies, and further research to develop screening strategies for endometrial cancer, so that patients can ensure accuracy while minimizing the pain and cost of the screening process.

  1. Thank you very much for your discussion on whether ECT technology is better than ultrasound technology for premenopausal female EC patients.

Since the original literature included in this study did not compare the diagnostic ability of the two technologies for EC, it is not possible to conclude that ECT is superior to ultrasound for EC in premenopausal women based on the current research results. We will keep an eye on this question, and further scientific discussion will be possible if further research results are available.

  1. Thank you for the discussion on metabonomics in the discussion.

The purpose of this part of the discussion is explained here: we are also concerned about the relatively low sensitivity of the current ECT technology, and the discussion part extends some of the shortcomings of ECT technology that may improve ECT technology (including AI technology to reduce the differences of pathologists, combined metabolomics analysis, etc.).

However, metabolomics is still in the preliminary stage of research, and it is still unknown whether a small amount of secretions in ECT swabs can detect enough substances to improve sensitivity with cytological tests.

Reply to Minor points:

  1. Thank you very much for your comments about grammar. We found a professional language retouching team to do the language retouching.
  2. Thank you very much for your suggestion on comparing the diagnostic effectiveness of different ECT tools. Due to the limitations of the published literature, it is not possible to make a meta-analysis of the research in this direction at this time, but we will keep an eye on it and further scientific discussion can be conducted if there are further research results.
  3. Thank you for your suggestions on the revision of the chart. First of all, it needs to be explained that a large number of indicators and pictures are integrated in this study, so we summarize the results in the table. The table was not intuitive enough, so we re-edited it to better highlight the key points.

Thank you again for taking time out of your busy schedule to read and review the draft we submitted.

Reviewer 2 Report

This is an extensive systematic review on the role of role of endometrial cytology in the diagnosis of endometrial cancer.

The subject is of clinical interest and the methods are in general adequate.

Of course, the selection process of the analyzed papers is of greatest importance for the results.

The authors state that they ruled out very early 8,930 papers (figure 1) as they were "not suitable for research purposes". Could the authors please provide sume more details about how they managed the analysis and differentiation of this large amount of papers and how they decided to exclude these papers from further analysis.

The discussion and interpretation of the presented results is written with caution which is adequate.

Author Response

First, thank you very much for your review comments. Regarding your questions, we provide the following feedback:

Due to the large volume of the excluded documents, it is inconvenient to show the specific process of the excluded documents in the manuscript. We keep Word manuscripts marked with the reasons for the specific types of excluded documents in the process of the excluded documents. If necessary, we can provide relevant documents to further explain the specific reasons for the excluded original documents.

Thank you again for taking time out of your busy schedule to read and review the draft we submitted.

PS:

SEARCH QUERY

('endometrial sampling':ab,ti OR 'endometrial cytology':ab,ti OR 'endometrial cytobrush':ab,ti OR 'endometrial cytotape':ab,ti OR 'endometrial samples collection':ab,ti OR 'endometrial cotton swab':ab,ti OR 'uterine lavage':ab,ti OR 'endometrial cells':ab,ti OR 'tao brush':ab,ti OR 'sap i':ab,ti OR pipelle:ab,ti OR 'endometrial sampling device for cytology':ab,ti) AND ('endometrial cancer':ab,ti OR 'endometrial carcinoma':ab,ti OR 'endometrial hyperplasia':ab,ti OR 'simple endometrial hyperplasia':ab,ti OR 'complex endometrial hyperplasia':ab,ti OR 'atypical endometrial hyperplasia':ab,ti OR 'endometrial intraepithelial neoplasia':ab,ti) AND (screening:ab,ti OR diagnosis:ab,ti OR diagnose:ab,ti)

Reviewer 3 Report

Dear authors,

It is an excellent manuscript, very interesting, and well prepared. I have no remarks, no improvements are needed regarding content of the manuscript. It can be published in its current content form. However, the paper needs extensive English editing. I suggest to review the paper with native speaker. There are many sentence constructions that need corrections (e.g. Because the incidence of endometrial cancer....; And a third researcher....; After the two researchers independently conducted..., and many more)

Author Response

First, thank you very much for your review comments. Regarding your questions, we provide the following feedback: We found a professional language retouching team to do the language retouching.

Thank you again for taking time out of your busy schedule to read and review the draft we submitted.

Round 2

Reviewer 1 Report

The authors did not try to address most of my major concerns from the previous submission. They did not include any more literature findings on pre vs post menopause or information on how the ETC compares to ultrasound diagnosis, or at least discussing these gaps in knowledge in the discusssion. In addition, they left in the metabolomics paragraph within the discussion without introducing it in the introduction or results sections. 

Author Response

REPLY

First, thank you very much for your review comments. Regarding your questions, we provide the following feedback:

As for the comparison of ECT techniques before and after menopause and the comparison of diagnostic effect between ultrasound examination and ECT, we have explained them in the discussion section of the following manuscript due to the limitation of the original literature (not enough literature was retrieved to support meta-analysis).

As for the content of metabolomics you proposed, our aim is to provide a possible scheme of joint detection for ECT. It is not relevant to the main content of this study, and has been deleted from the revised manuscript.

Thank you again for taking time out of your busy schedule to read and review the draft we submitted.
